# Extracellular Vesicle-Based Bronchoalveolar Lavage Fluid Liquid Biopsy for EGFR Mutation Testing in Advanced Non-Squamous NSCLC

**DOI:** 10.3390/cancers14112744

**Published:** 2022-05-31

**Authors:** In Ae Kim, Jae Young Hur, Hee Joung Kim, Wan Seop Kim, Kye Young Lee

**Affiliations:** 1Precision Medicine Lung Cancer Center, Konkuk University Medical Center, Seoul 05030, Korea; 20180618@kuh.ac.kr (I.A.K.); 20160475@kuh.ac.kr (J.Y.H.); hjkim@kuh.ac.kr (H.J.K.); wskim@kuh.ac.kr (W.S.K.); 2Department of Pulmonary Medicine, Konkuk University School of Medicine, Seoul 05030, Korea; 3Department of Pathology, Konkuk University School of Medicine, Seoul 05030, Korea; 4Exosignal, Inc., Seoul 05030, Korea

**Keywords:** bronchoalveolar lavage (BAL), liquid biopsy, extracellular vesicles, EGFR mutation testing, NSCLC

## Abstract

**Simple Summary:**

Tissue biopsy is the gold standard for molecular genotyping in lung cancer. However, obtaining tumor tissue is challenging due to its invasiveness, inadequate amount of tissue, or complications. To overcome the limitations of tissue biopsy, plasma liquid biopsy using cfDNA has been investigated extensively; however, its low sensitivity limits the clinical application. Therefore, we used the tumor-specific DNA of extracellular vesicles (EVs) in bronchoalveolar lavage fluid (BALF) as DNA source for EGFR genotyping. As a result, we demonstrated that EV-based BALF EGFR testing in advanced lung NSCLC is a highly accurate rapid method overcoming low sensitivity of plasma cfDNA-based EGFR genotyping. It can be used as an adjuvant or alternative method for lung biopsy in cases where obtaining an adequate amount of tissue is difficult.

**Abstract:**

To overcome the limitations of the tissue biopsy and plasma cfDNA liquid biopsy, we performed the EV-based BALF liquid biopsy of 224 newly diagnosed stage III-IV NSCLC patients and compared it with tissue genotyping and 110 plasma liquid biopsies. Isolation of EVs from BALF was performed by ultracentrifugation. EGFR genotyping was performed through peptide nucleic acid clamping-assisted fluorescence melting curve analysis. Compared with tissue-based genotyping, BALF liquid biopsy demonstrated a sensitivity, specificity, and concordance rates of 97.8%, 96.9%, and 97.7%, respectively. The performance of BALF liquid biopsy was almost identical to that of standard tissue-based genotyping. In contrast, plasma cfDNA-based liquid biopsy (*n* = 110) demonstrated sensitivity, specificity, and concordance rates of 48.5%, 86.3%, and 63.6%, respectively. The mean turn-around time of BALF liquid biopsy was significantly shorter (2.6 days) than that of tissue-based genotyping (13.9 days; *p* < 0.001). Therefore, the use of EV-based BALF shortens the time for confirmation of EGFR mutation status for starting EGFR-TKI treatment and can hence potentially improve clinical outcomes. As a result, we suggest that EV-based BALF EGFR testing in advanced lung NSCLC is a highly accurate rapid method and can be used as an alternative method for lung tissue biopsy.

## 1. Introduction

Testing for epidermal growth factor receptor (EGFR) mutations is an essential step during therapeutic decision-making for patients newly diagnosed with advanced non-small cell lung cancer (NSCLC), especially considering the remarkable clinical outcomes of EGFR-tyrosine kinase inhibitor (TKI) treatments [1]. The most common sensitive EGFR mutations are exon 19 deletion and 21L959R. EGFR-TKIs treatment with the sensitive EGFR mutations have demonstrated significantly improved progression-free survival rate and objective response rate, compared with chemotherapy [2,3,4,5].

For testing EGFR mutations, using DNA extracted from tumor tissues is the gold standard. However, obtaining tumor tissue samples is challenging, as tumors can be located at sites that are difficult to approach using percutaneous biopsy needles; moreover, a scarce amount of tumor tissue is obtained that is insufficient for genetic analysis. Additionally, this method is associated with a high possibility of complications, such as hemorrhage or pneumothorax [6]. Furthermore, it can take up to 2–3 weeks to confirm EGFR mutation status through conventional tissue biopsy.

Compared with tissue biopsy, liquid biopsy is repeatable, relatively less invasive, and less time-consuming. Liquid biopsy using cell-free DNA (cfDNA) has been extensively investigated in the context of both diagnostic and prognostic biomarkers for lung cancer [7,8]. CfDNA in the plasma has a high specificity in detecting EGFR mutations but is associated with a low sensitivity; this is because very small amounts of cfDNA exist in the blood as cfDNA is released from tumor cells only during cell death and is rapidly cleared from the circulation [9]. Thus, the application of liquid biopsy using cfDNA in clinical practice is limited [10,11]. Extracellular vesicles (EVs) are heterogeneous, nano-sized membrane vesicles that serve as key messengers of intracellular communication. EVs are ideal carriers of cancer biomarkers as cancer cells secrete abundant EVs, the contents of which reflect the molecular and genetic composition of the parental cancer cells [12,13]. These nano-sized vesicles contain many molecules, such as nucleic acids, including tumor-specific oncogenic mutant DNA and various subtypes of RNA, proteins, and lipids, and are involved in tumor proliferation, progression, and metastasis [14]. Moreover, the lipid bilayer of EVs allows for stable cargoes that are relatively difficult to degrade. These characteristics would allow for higher sensitivity of mutation detection, making EVs ideal candidates for liquid biopsy. However, research on EGFR genotyping using EV-derived DNA in the plasma is limited. Previous studies have reported that the combined use of EV-derived RNA and cfDNA in the plasma improves the sensitivity of EGFR mutation detection by 88–92% [5,15]; however, to the best of our knowledge, the use of EV-derived DNA (EV-DNA) alone for this purpose has not been reported.

There are several research works on the EV-based EGFR mutation detection [16,17,18]. To increase the sensitivity in plasma EGFR mutation testing, the method combining the cfDNA with DNA and RNA derived from exosomes are used [19,20]. We also previously reported that EVs isolated from bronchoalveolar lavage fluid (BALF) of patients with NSCLC contain abundant double-stranded DNA and both the detection and concordance rate of EGFR mutations using BALF-EVs increase with tumor stages [21,22]. The results suggest that the shedding of tumor-specific EVs containing mutant EGFR DNA is proportional to the tumor burden. Although EGFR genotyping using EV-based BALF liquid biopsy was associated with a low sensitivity in stage I NSCLC, it was successful for advanced NSCLC that requires urgent and accurate EGFR mutation testing decisions. Therefore, we conducted a large-scale prospective study to validate the clinical utility of EGFR mutation testing using EV-based BALF liquid biopsy for patients with advanced non-squamous NSCLC.

## 2. Materials and Methods

### 2.1. Patients and Sample Collection

Paired tissue and BALF of 224 patients with lung cancer from June 2017 to August 2020 were prospectively collected at our hospital and included in this study. Patients with suspected lung cancer based on chest tomography (CT) underwent bronchoscopy for biopsy during initial lung cancer workup. BALF was obtained from the sub-segmental bronchus site where the tumor mass was located. Bronchoscopy was performed and at least 10 mL of BALF was collected by aspiration after instillation of approximately 50–70 mL of sterile isotonic saline by wedging the bronchoscope at the segment where the tumor was located. BALF was collected as a residue after samples were submitted for routine cytological examination and microbial study. Additionally, 110 of the 224 patients had matched blood samples collected on the same day of bronchoscopy. To confirm diagnosis, we obtained tumor tissue by endobronchial biopsy for tumors with visible lesions. For tumors without visible lesions, we performed blind trans bronchial lung biopsy (TBLB), CT-guided percutaneous needle biopsy (PCNB), or surgical lung resection. Tumor staging was based on the 8th TNM classification criteria [23]; clinical data of the enrolled patients were reviewed by medical records. Patient demographics and clinical characteristics are detailed in Table 1. This study was conducted in accordance with the Declaration of Helsinki (2013) and the study protocol was approved by the institutional review board of Konkuk University Medical Center (KUH 1010868). Written informed consent was obtained from all patients.

### 2.2. BALF Processing and EV Isolation

EVs were isolated from 1 mL of BALF samples within 2 h of collection. Briefly, cells and debris were removed by centrifugation at 1000 g for 10 min at 4 °C. Next, the cell- and debris-free BALF sample was spun in an ultracentrifuge tube at 200,000 g for 1 h at 4 °C using a Beckman rotor (Beckman Coulter, Brea, CA, USA). The supernatant was carefully removed and discarded, and the pellet was suspended in 200 μL of phosphate-buffered saline. The EV pellet was then lysed by mixing lysis buffer (10 mM Tris-HCl, 20% Triton X-100) and detergent to isolate EV-derived DNA, which was further purified using the High Pure PCR Template Preparation Kit (Roche Diagnostics, Mannheim, Germany). The quality and length of the purified DNA were analyzed using a 4200 Tapestion and Genomic DNA ScreenTape (Agilent Technologies, Santa Clara, CA, USA). The concentration and purity of DNA samples were measured using the NanoDrop (Thermo Scientific, Waltham, MA, USA).

### 2.3. EV-Based BALF EGFR Genotyping

The PANAMutyper™ R EGFR kit (Panagene, Daejeon, Korea) was used for detecting EGFR mutations. The PANAMutyper™ R EGFR kit (PANA C-Melting™) combines peptide nucleic acid (PNA)-based PCR clamping (PNAClamp™) [24] with multiplex fluorescence melting curve analysis (PANA S-Melting™) using fluorescence-labeled PNA probes that allow detection of 47 hotspot mutations in EGFR exons 18–21. PCR was performed with a total reaction volume of 25 μL containing 70 ng template DNA, primer and PNA probe sets, along with the master mix using the CFX96 real-time PCR detection system (Bio-Rad, Hercules, CA, USA). A detection probe was designed for competitive hybridization with clamping PNA on the same strand (either sense or antisense DNA strand). The PCR was performed with two holding periods of 50 °C for 2 min and 95 °C for 15 min and then (i) 15 cycles of 95 °C for 30 s, 70 °C for 20 s, and 63 °C for 60 s; and (ii) 35 cycles of 95 °C for 10 s, 53 °C for 20 s, and 73 °C for 20 s. Fluorescence was measured on all four channels (FAM, ROX, Cy5, and HEX) during PCR and melting curve analysis. Each sample was then genotyped based on the melting temperature (Tm) that was determined from the melting peak of each fluorescent dye [25]. In summary, two specifically designed PNA oligomers, a clamping PNA, which suppresses the amplification of an undesired or wild-type allele, and PNA detection probe, which has a fluorophore and a quencher group at each terminus of the probe were used in this qPCR.

### 2.4. EGFR Genotyping of Tissue DNA and Plasma cfDNA

Formalin-fixed, paraffin-embedded (FFPE) tissues were prepared; tumor DNA was extracted and purified using the TANBead OptiPure FFPE DNA Tube (Taiwan Advanced Nanotech, Taoyuan, Taiwan). EGFR genotyping was then performed as described above.

Blood samples (5 mL) were collected in K2-EDTA tubes; plasma was isolated via centrifugation at 1000 g at 4 °C for 10 min within 2 h of collection. Plasma cfDNA was isolated and purified using either the High Pure PCR Template Preparation Kit (Roche Diagnostics) [26] or TANBead OptiPure cfDNA Auto Tube (Taiwan Advanced Nanotech). EGFR genotyping was then performed as described above. The quality and length of the purified DNA were analyzed using a 4200 Tapestion and Genomic DNA ScreenTape (Agilent Technologies, Santa Clara, CA, USA). The concentration and purity of DNA samples were measured using the NanoDrop (Thermo Scientific, Waltham, MA, USA).

### 2.5. Statistical Analysis

EGFR status assessment in the histological tumor samples was considered as the standard reference for calculation of concordance, sensitivity, and specificity of BALF and plasma cfDNA EGFR mutation detection. Categorical variables were summarized by calculating frequencies and percentages. Means and standard deviations were used to determine numerical variables. The Spearman’s correlation test was used for correlation analysis and Pearson chi-square and Fisher’s exact tests were used to determine the significance of differences in EGFR mutation rate, sensitivity trends of BALF liquid EGFR genotyping, and comparison with clinical parameters. The concordance rate was calculated as the sum of positives and negatives in samples divided by the total number of matched samples. Sensitivity was calculated as the proportion of concordant positives in samples out of the positive tissue samples, whereas specificity was calculated as the proportion of concordant negatives in samples out of the negative tissue samples. All statistical analyses were carried out using SPSS version 25.0 (IBM Corp, Armonk, NY, USA), and a *p*-value < 0.05 was considered statistically significant.

## 3. Results

### 3.1. Patient Characteristics

Of the 224 newly diagnosed patients with NSCLC included in the present study, 40.2% were female, 47.3% were non-smokers, and the majority (83.5%) were histologically classified as having adenocarcinoma. EGFR-mutant cases were identified in 93 patients (41.5%) via tissue-based genotyping, whereas the remaining 131 patients had wild-type EGFR (Table 1). Of the 93 patients with EGFR mutations, 52 (55.9%) showed an exon 19 deletion, 33 (35.4%) showed an exon 21 p.L858R mutation, and 6 (2.6%) had double EGFR mutations. Additionally, 16.1%, 43.3%, and 40.6% had stage III, IVA, and IVB NSCLC, respectively.

We also performed EGFR genotyping using plasma cfDNA from 110 patients; the sex ratio and proportion of smokers were similar in the paired tissue, BALF, and plasma subgroups. We did not observe significant demographic differences between the total cohort and the plasma subgroup (Table 1).

### 3.2. Methods of Tissue Procurement for EGFR Genotyping

Visible endobronchial lesions were present in 20.1% of the total patients, and tissue was obtained without difficulty by bronchoscopy. However, the remaining 79.9% did not have visible endobronchial lesions, and hence transbronchial needle aspiration (TBNA), PCNB, or surgical biopsy was performed to obtain adequate tumor tissue. Endobronchial ultrasonography-TBNA for mediastinal lymph nodes was performed in 35.2% of the patients, although the procedure is time-consuming, involves a high cost, and requires advanced techniques and apparatus. CT-guided PCNB, which is associated with a risk of pneumothorax and hemorrhage, was performed in 19.2% of the patients. Blind transbronchial lung biopsy was performed in 9.8% of the patients as the lesions were hidden and forceps could not be guided exactly to the right site. Surgical resection of the lung for biopsy under general anesthesia was performed in 8.9% of the patients, whereas cytology of pleural effusion or sputum was used in 6.7% of the patients for lung cancer diagnosis and EGFR testing (Table 2). These data highlight the difficulties, time, effort, and cost involved in obtaining lung tissue samples.

### 3.3. EGFR Mutation Detection Rate in Liquid Biopsy Using BALF and Plasma and Tissue Samples

The sensitivity, specificity, positive predictive value (PPV), negative predictive value (NPV), and concordance rate for BALF liquid EGFR genotyping were compared with those for tissue-based EGFR genotyping and were 97.8%, 97.7%, 96.8%, 98.4%, and 97.7%, respectively. The false negative rate was 2.1% (2/93), whereas the false positive rate was 2.3% (3/131), suggesting that the performance of BALF liquid EGFR genotyping was comparable to that of tissue-based EGFR genotyping. Similarly, the sensitivity, specificity, PPV, NPV, and concordance rate for plasma cfDNA-based liquid biopsy (*n* = 110) were compared with those of tissue-based EGFR genotyping and were 48.5%, 86.3%, 84.2%, 52.7%, and 63.6%, respectively. The sensitivity, specificity, and concordance rate for plasma cfDNA-based liquid biopsy in comparison to those of tissue-based EGFR genotyping were markedly lower than those of BALF liquid EGFR genotyping (Table 3).

### 3.4. Concordance Rate for EGFR Genotyping among Tissue Biopsy, BALF, and Plasma Liquid Biopsy (n = 110)

Next, we compared the concordance rates of EGFR genotyping in matched tissues, BALF, and plasma of the 110 patients. The concordance rate between BALF and tissue was 99.1%, whereas the concordance rate between tissue and plasma cfDNA was only 63.6%. (Figure 1). The false positive rate of plasma liquid biopsy was 13.6% (6/44) whereas the false positive rate of BALF liquid biopsy was 0%. The false negative rate of plasma liquid biopsy was 51.5% (34/66) while the false negative of BALF liquid biopsy was 1.5% (1/66) (Appendix A). These results suggest that BALF liquid EGFR genotyping is more sensitive and accurate than plasma cfDNA-based EGFR genotyping.

### 3.5. Comparison of Sensitivity between BALF and Plasma Liquid Biopsy Depending on Presence of Metastasis

We compared the sensitivity of EGFR mutation testing using plasma and BALF depending on the tumor stage in Figure 2. The tumor stages are classified with the type of metastasis [23]. In plasma cfDNA-based liquid biopsy, the sensitivity increased with the extent of the disease and was 19.2%, 40%, and 62.5% in stages M1a, M1b, and M1c, respectively. Noticeably, the sensitivity of plasma cfDNA-based liquid biopsy in stage IVB associated with extensive metastatic spread was only 62.5%. The sensitivity of plasma cfDNA-based liquid biopsy was only 19–33% in intrathoracic disease where cancer is confined to the thorax and approximately 40–62% in extrathoracic cases associated with extensive spread. In contrast, the sensitivity of BALF liquid biopsy for EGFR genotyping was nearly 100%, regardless of the extent of the metastasis (Figure 2). In other words, the BALF liquid biopsy was superior to cf DNA test in the aspect of detection of EGFR mutation in intrathoracic metastasis (100% vs. 19.2 %, *p* < 0.001).

### 3.6. Turnaround Time of EGFR Mutation Testing in BALF-Based and Tissue-Based Genotyping

The turn-around time of tissue/cytology-based EGFR mutation testing was approximately 13.9 days at Konkuk University Medical Center (95% CI, 9.4–15.5 days). However, compared to tissue-based genotyping, BALF liquid EGFR testing had a significantly lower turnaround time (approximately 2.6 days; 95% CI, 1.8–2.9 days; *p* < 0.001) (Table 4), suggesting that BALF liquid EGFR testing is a rapid method for assessing EGFR mutation status and can lead to efficient treatment decision-making. The time-consuming paraffin fixation is required for tissue-based EGFR mutation testing. Using fresh tissue samples to save paraffin fixation time is not realistic or appropriate for advanced lung cancer patients, considering that only a very small amount of sample can be obtained from them. BALF EGFR mutation testing does not require paraffin fixation time of tissue so it can save the time for EGFR mutation testing.

## 4. Discussion

Tissue biopsy is the gold standard for molecular genotyping in lung cancer; however, it is an invasive procedure that often causes complications. To circumvent the limitations of tissue biopsy, such as inadequate sample and invasiveness, liquid biopsy using plasma cfDNA has been investigated extensively. However, the low sensitivity of liquid biopsy due to the short half-life and intrinsic instability of plasma cfDNA poses a challenge in its clinical application. BALF has emerged as an alternative liquid biopsy source with high sensitivity, because it contains cellular and non-cellular components directly released by tumor cells and the tumor microenvironment. EVs isolated from BALF contain double-stranded DNA [21,27] and can thus be utilized as an alternative to tumor tissue for EGFR genotyping [28]. EVs are 30–200-nm-sized nanoparticles enclosed by bilayer lipid membranes, which carry RNA, DNA, proteins, lipids, and other diverse bioactive molecules and are involved in intercellular communication [29]. EVs are actively released by cancer cells and are produced in greater quantities by cancer cells than normal cells; EV production increases with the extent of the disease [30]. EV-DNA presents several advantages over plasma cfDNA, as EV biogenesis is an active process in tumor cells that reflects the state of tumor progression and DNA encapsulated by the lipid bilayer of EVs is extremely stable and protected from degradation by external factors, unlike cfDNA, which is fragmented, free-floating, and a product of apoptosis [31,32]. Additionally, recent next-generation sequencing studies have demonstrated that EV-DNA can serve as good reserves of cancer biomarkers [33,34,35]. These findings highlight the potential application of EV-based liquid biopsy for EGFR genotyping in patients with advanced NSCLC.

Previously, we reported on the usefulness of EV-based BALF liquid biopsy for EGFR genotyping in patients with NSCLC [22]. The sensitivity and specificity of BALF EV-based EGFR genotyping in all stages were 76% and 87%, respectively, and the sensitivity was increased significantly according to each TNM stage reaching to 100% in stage IV metastatic NSCLC patients [26]. However, the clinical application of this novel platform was limited by the small sample size of only 95 advanced NSCLC cases. Thus, in the present study, we aimed to validate EV-based BALF EGFR genotyping prospectively for a large sample size in a real clinical setting of 224 patients with advanced non-squamous NSCLC who need urgent therapeutic intervention. Herein, the sensitivity of BALF liquid EGFR testing was 96.8–100% regardless of the presence of extrathoracic metastasis, while the sensitivity of plasma liquid biopsy was 19.2–33% in cases with intrathoracic metastasis and only 40.0–62.5% even in cases with extrathoracic metastasis. The concordance rate between BALF- and tissue-based EGFR genotyping was 98.5%, compared to 43.9% between plasma cfDNA- and tissue-based EGFR genotyping, highlighting the potential of BALF liquid EGFR testing as a replacement for conventional tissue-based genotyping, at least in patients with advanced non-squamous NSCLC. Additionally, EGFR status was rapidly assessed via BALF liquid biopsy at least 10 days earlier than with conventional tissue-based genotyping. Considering the immediate symptomatic improvement with the correct EGFR-TKI treatment, the ultra-rapid turnaround time of BALF liquid biopsy is potentially beneficial for advanced NSCLC patients who need urgent therapeutic intervention, especially symptomatic or critically ill patients. Currently, it might be challenging to prescribe the anticancer agent based only on oncogenic mutants without histologic confirmation; however, given its high sensitivity and specificity, liquid biopsy can be reasonably and feasibly used to guide therapeutic decision. We expect that this study will contribute to support the paradigm shift from current histology-based lung cancer diagnosis to genetic or molecular liquid biopsy-based lung cancer diagnosis.

Ground glass opacity, air bronchogram, pleural retraction, and vascular convergence are significantly more prevalent in EGFR-mutant NSCLC [36] and are factors that negatively affect safe tissue biopsy. Hence, surgical biopsy is infrequently recommended in such cases. Additionally, waiting for enough tumor growth to target for tissue biopsy can lead to a diagnostic delay. Active mutations in the EGFR-tyrosine kinase domain, such as exon 19 deletion and exon 21L858R, are found specifically in lung cancer and are not common in other cancers; for example, the frequency of EGFR mutations in colon cancer is 3% and lung-cancer-specific mutations are rare [37]. Therefore, detection of lung-cancer-specific mutant EGFR DNA in BALF along with compatible CT findings suggesting malignancy may be sufficient to diagnose EGFR-mutant lung cancer to guide EGFR-TKI treatment. In East Asian countries, the prevalence of EGFR-mutant NSCLC is 30–40% and as high as 65% in females, never smokers, and minimally exposed ex-smokers [38]. Bronchoscopy is one of the basic procedures for initial diagnostic workup in patients with suspected malignancy. The procedure is inexpensive and BALF can be safely collected during bronchoscopy. Therefore, BALF liquid EGFR testing has various advantages, especially in patients with EGFR mutation-associated factors, such as East Asian ethnicity, female, never smoker or minimally exposed ex-smoker status, and peripheral tumors, allowing for ultra-rapid and accurate diagnosis that can guide treatment. Therefore, further validation and development of BALF liquid biopsy for use in the clinical setting is required.

This study had some limitations. First, it was a single-center study; hence, multi-center, large-scale prospective clinical trials are required to validate our results. Second, standardization of EV isolation and EV-DNA extraction is required for application in routine clinical practice. Although we used ultracentrifugation, which is one of the standard methods for EV isolation, it might be difficult to apply in routine practice, necessitating the development of a one-step kit. Additionally, BALF sample collection and storage should be standardized. Third, technical aspects of the BAL procedure, such as contamination with blood due to touch bleeding, amount of infused saline, and BALF retrieval efficiency due to the patient’s position and tumor location, especially in the upper lobe posterior segment and lower lobe superior segment, might affect BALF liquid biopsy outcomes. However, in our experience, tumor location does not necessarily affect BALF liquid biopsy results in advanced NSCLC, as peri-tumoral fibrosis results in easy retrieval of lavage fluids due to the applied negative pressure, while retrieval efficiency in normal compliant lung is usually decreased due to collapse.

## 5. Conclusions

We demonstrated that EV-based BALF EGFR testing in advanced lung NSCLC is a highly accurate and rapid method, which can overcome the low sensitivity of plasma cfDNA-based EGFR genotyping and can be used as an adjuvant or alternative method for lung biopsy in cases where obtaining an adequate amount of tissue is difficult. Additionally, the use of EV-based BALF may shorten the time for confirmation of EGFR mutation status for starting EGFR-TKI treatment and can hence potentially improve clinical outcomes.

## Figures and Tables

**Figure 1 cancers-14-02744-f001:**
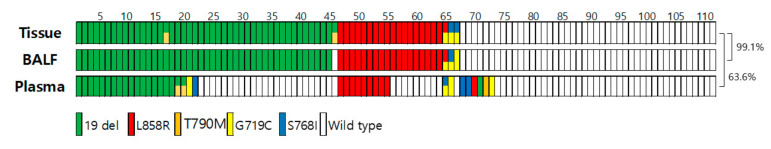
Concordance rate for EGFR genotyping among tissue biopsy, BALF liquid and plasma liquid biopsy (*n* = 110) (double colors mean double EGFR mutation).

**Figure 2 cancers-14-02744-f002:**
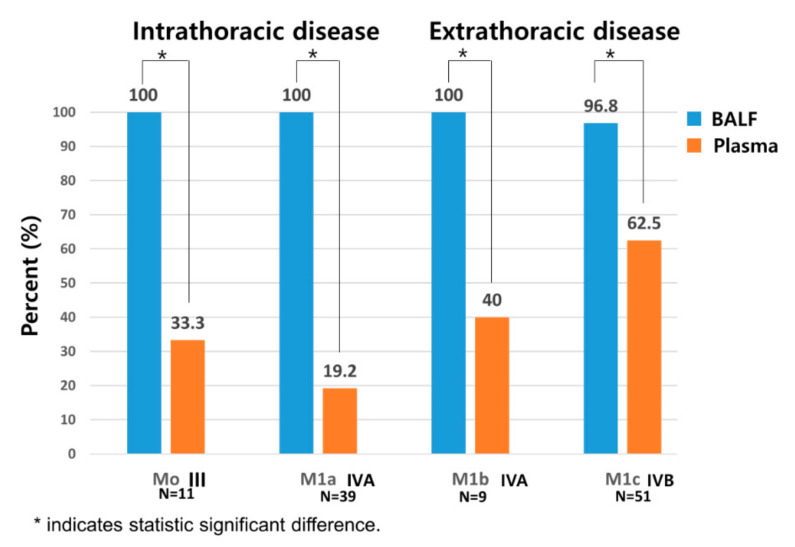
Comparison of sensitivity between BALF and plasma liquid biopsy depending on metastatic tumor stage. * Indicates statistically significant difference (*p* < 0.05). (M0: no metastasis; M1a: intrathoracic metastasis; M1b: single extrathoracic metastasis; M1c: multiple extrathoracic metastasis).

**Table 1 cancers-14-02744-t001:** Patient demographics and clinical characteristics.

Characteristics	BALF, n (%)	Plasma, n (%)	*p*-Value
**Number of patients**	224	110	
**Age** (mean ± SD)	67.7 ± 11.8	67.4 ± 10.1	0.82
**Sex**			
Male	134 (59.8)	53 (48.2)	0.19
Female	90 (40.2)	57 (51.8)	
**Smoking history**			
Non-smoker	106 (47.3)	61 (55.5)	0.38
Ex-smoker	58 (25.9)	24 (21.8)	
Current smoker	60 (26.8)	25 (22.7)	
**Histology**			
Adenocarcinoma	187 (83.5)	95 (86.4)	0.52
NSCLC	37 (16.5)	15 (13.6)	
Stage			
III	36 (16.1)	11 (10)	0.18
IVA	97 (43.3)	48 (43.6)	
IVB	91 (40.6)	51 (46.4)	
**Tissue EGFR mutation**			
Wild type	131 (58.5)	44 (40)	0.12
EGFR mutation	93 (41.5)	66 (60)	
Exon 19 del	52 (55.9)	43 (39.1)	
L858R	33 (35.4)	18 (16.4)	
Exon 19 del + T790M	2 (2)	2 (1.8)	
G719C + S768I	2 (2)	2 (1.8)	
L858R + G719C	1 (1)	1 (0.9)	
L858R + T790M	1 (1)	0 (0)	
G719C	1 (1)	0 (0)	
L861Q	1 (1)	0 (0)	
S768I	0 (0)	0 (0)	
T790M	0 (0)	0 (0)	

Abbreviations: BALF, bronchoalveolar lavage fluid; del, deletion; EGFR, epidermal growth factor receptor; NSCLC, non-small cell lung cancer.

**Table 2 cancers-14-02744-t002:** Methods for tissue-based EGFR genotyping.

Biopsy Methods	*n*	%
**Endobronchial lesion (+)**	45	20.1
Endobronchial biopsy	29	12.9
Endobronchial biopsy + EBUS	15	6.7
Negative endobronchial biopsy + PCNB	1	0.4
**Endobronchial lesion (−)**	179	79.9
EBUS-TBNA	79	35.2
CT-guided PCNB	43	19.2
TBLB	22	9.8
Surgical resection	20	8.9
Cytology *	15	6.7

Endobronchial lesion (+) is defined as the existence of endobronchial nodules. Endobronchial lesion (−) is defined as the absence of endobronchial lesions. * Cytology samples (pleural effusion, 13; sputum, 2) Abbreviations: CT, computed tomography; EBUS, endobronchial ultrasonography; PCNB, percutaneous needle biopsy; TBLB, transbronchial lung biopsy; TBNA, transbronchial needle aspiration.

**Table 3 cancers-14-02744-t003:** Comparison of EGFR mutation detection rate in liquid biopsy between BALF and plasma.

EGFR Genotype	Tissue	BALF (*n* = 224)		Tissue	Plasma (*n* = 110)	
		Mutant	Wild Type		Mutant	Wild Type
**Mutant type**	93	91	2	66	32	34
**Wild type**	131	3	128	44	6	38
**Sensitivity**	97.8% (91/93)	(95% CI, 92.4–99.7)		48.5% (32/66)	(95% CI, 35.9–61.1)	
**Specificity**	97.7% (128/131)	(95% CI, 93.5–99.5)		86.3% (38/44)	(95% CI, 72.6–94.8)	
**PPV**	96.8% (91/94)	(95% CI, 90.8–98.9)		84.2% (32/38)	(95% CI, 70.8–92.1)	
**NPV**	98.4% (128/130)	(95% CI, 93.5–99.5)		52.7% (38/72)	(95% CI, 46.2–72.6)	
**Concordance rate**	97.7% ((91 + 128)/224) (95% CI, 94.8–99.2)			63.6% ((32 + 38)/110) (95% CI, 53.9–72.6)		

Abbreviations: BALF, bronchoalveolar lavage fluid; CI, confidence interval; EGFR, epidermal growth factor receptor; PPV, positive predictive value; NPV, negative predictive value.

**Table 4 cancers-14-02744-t004:** Comparison of the turnaround time of EGFR mutation testing using BALF vs. tissue samples.

Sample Type	Mean (Days)	Median (Days)	*p*-Value
**BALF**	2.6 ± 2.03	2	<0.001
**Tissue**	13.9 ± 12.4	12	

## Data Availability

The data presented in this study are available from the corresponding author upon reasonable request.

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
