# Peer review of "Extracellular Vesicle-Based Bronchoalveolar Lavage Fluid Liquid Biopsy for EGFR Mutation Testing in Advanced Non-Squamous NSCLC"

_cancers, 2022, doi:10.3390/cancers14112744_

Round 1

Reviewer 1 Report

  1. The introduction of the article should be modified. Authors should cite similar articles with preliminary results on EV-based EGFR mutation detection in lung cancer. For Ex (Wan. Y et al; Annals of oncology, 29: 2379-2383; 2018, Purcell. E et al, Front Cell Dev Biol, 2021.; Park. J et al, Cancers, 2020)

Author Response

The introduction of the article should be modified. Authors should cite similar articles with preliminary results on EV-based EGFR mutation detection in lung cancer. For Ex (Wan. Y et al; Annals of oncology, 29: 2379-2383; 2018, Purcell. E et al, Front Cell Dev Biol, 2021.; Park. J et al, Cancers, 2020)

A: We cited similar articles with preliminary results on EV-based EGFR mutation detection in lung cancer including the references suggested by the reviewer.

 In line 79-81, “There are several research works on the EV based EGFR mutation detection [16-18]. To increase the sensitivity in plasma EGFR mutation testing, the method combining the cfDNA with DNA and RNA derived from exosome are used [19,20].” was added.

Reviewer 2 Report

The manuscript entitled "Extracellular vesicle-based bronchoalveolar lavage fluid liquid biopsy for EGFR mutation testing in advanced non-squamous NSCLC" presents the results of the study comparing the sensitivity, specificity, and concordance rate of tissue-based EGFR genotyping with EGFR genotyping in extracellular vesicles (isolated from bronchoalveolar lavage fluid) and cfDNA (from serum). It has been shown that the BALF liquid biopsy demonstrated very high sensitivity, specificity, and concordance rate (97.8%, 96.9%, and 97.7%, respectively) in comparison to the standard tissue-based genotyping, whereas the cell-free DNA genotyping gave much lover results, particularly related to the sensitivity and concordance rate.

The study is well planned and manuscript is well written, limitations of the study are accurately described. The introduction provide sufficient background, however it would be good to add paragraph regarding the most popular EGRF mutations in the tested population and their clinical implications.

More information regarding the method of EGFR genotyping (peptide nucleic acid clamping-assisted fluorescence melting curve analysis) could be provided in the method section,, 

In the result section - 

In section 3.3 there is used the "BALiquid EGFR"  - is it an abbreviation ?

In the Table 3 there is problem with determinants of a decimal number - the "." or "," symbols are missing in some of the numbers (percentage number).

The concordance rate value regarding the cell-free DNA EGFR genotyping in  Table 3 is different from data in paragraph 3.4.

In the paragraph 3.5 no information on stage is given (and the paragph title relates to the tumor stage) - maybe it would be better to tunderline that this analysis is related to the presence of metastasis? The definition of the M0, M1a, M1b, M1c should be given.  

The autorhs show that the mean turnaround time of BALF liquid biopsy is significantly shorter (2.6 days) than that of tissue-based genotyping (13.9 days). The length of the procedure is due to the fact that paraffin fixed samples are used for genomic analysis. Would the EGFR mutation analysis direct in fresh samples without fixation (tissue sections from the tumor and from the surgical margin) accelerate the genotyping procedure?

Author Response

Reviewer 2

The manuscript entitled "Extracellular vesicle-based bronchoalveolar lavage fluid liquid biopsy for EGFR mutation testing in advanced non-squamous NSCLC" presents the results of the study comparing the sensitivity, specificity, and concordance rate of tissue-based EGFR genotyping with EGFR genotyping in extracellular vesicles (isolated from bronchoalveolar lavage fluid) and cfDNA (from serum). It has been shown that the BALF liquid biopsy demonstrated very high sensitivity, specificity, and concordance rate (97.8%, 96.9%, and 97.7%, respectively) in comparison to the standard tissue-based genotyping, whereas the cell-free DNA genotyping gave much lover results, particularly related to the sensitivity and concordance rate.

  • The study is well planned, and manuscript is well written, limitations of the study are accurately described. The introduction provides sufficient background, however it would be good to add paragraph regarding the most popular EGRF mutations in the tested population and their clinical implications.

A: We explained more about common EGFR mutation and clinical implication, as the reviewer suggested.

In line 47-50, “The most common sensitive EGFR mutations are exon 19 deletion and 21L959R. EGFR-TKIs treatment with the sensitive EGFR mutations have demonstrated significantly improved progression free survival rate and objective response rate, compared with chemotherapy [2-5].” was added

  • More information regarding the method of EGFR genotyping (peptide nucleic acid clamping-assisted fluorescence melting curve analysis) could be provided in the method section.

A: We provided more explanation about the method of EGFR genotyping as the reviewer recommended.

In line 133-136 “A detection probe was designed for competitive hybridization with clamping PNA on the same strand (either sense or antisense DNA strand). The PCR was performed with two holding periods of 50 °C for 2 min and 95 °C for 15 min and then (i) 15 cycles of 95 °C for 30 s, 70 °C for 20 s, and 63 °C for 60 s; and (ii) 35 cycles of 95 °C for 10 s, 53 °C for 20 s, and 73 °C for 20 s.” was added, and in line 140-143 “In summary, two specifically designed PNA oligomers, a clamping PNA, which suppresses the amplification of an undesired or wild-type allele, and PNA detection probe, which has a fluorophore and a quencher group at each terminus of the probe were used in this qPCR” was added

  • In the result section - In section 3.3 there is used the "BALiquid EGFR"  - is it an abbreviation ?

A: It was an abbreviation. To avoid any confusion, we replaced the previous “BALiquid EGFR” to “BALF liquid EGFR” in the revised manuscript.

  • In the Table 3 there is a problem with determinants of a decimal number - the "." or "," symbols are missing in some of the numbers (percentage number).

A: We fixed the errors in the table 3. The numbers are correctly written in the revised manuscript.

  • The concordance rate value regarding the cell-free DNA EGFR genotyping in Table 3 is different from data in paragraph 3.4.

A: The number in the table is correct number. We corrected the incorrect number 61.8% to 63.6% that agrees with that in Table 3.

  • In the paragraph 3.5 no information on stage is given (and the paragraph title relates to the tumor stage) - maybe it would be better to underline that this analysis is related to the presence of metastasis? The definition of the M0, M1a, M1b, M1c should be given.  

A: The definition on the M0, M1a, M1b, M1c was provided in figure 2 caption. We underlined this analysis is related to the presence of metastasis.

The following information is added to the figure 2 caption “M0: no metastasis, M1a: intrathoracic metastasis, M1b: single extrathoracic metastasis, M1c : multiple extrathoracic metastasis”

“The tumor stages are classified with the presence of metastasis [23].” was inserted at line 244 in page 8 after “We compared the sensitivity of EGFR mutation testing using plasma and BALF depending on the metastatic stage in Fig. 2.”

  • The authors show that the mean turnaround time of BALF liquid biopsy is significantly shorter (2.6 days) than that of tissue-based genotyping (13.9 days). The length of the procedure is due to the fact that paraffin fixed samples are used for genomic analysis. Would the EGFR mutation analysis direct in fresh samples without fixation (tissue sections from the tumor and from the surgical margin) accelerate the genotyping procedure?

A: We discussed the issue that the reviewer suggested in the revised manuscript. In line 265-270, “The time-consuming paraffin fixation is required for tissue-based EGFR mutation testing. Using fresh tissue samples to save paraffin fixation time is not realistic or appropriate for advanced lung cancer patients, considering that only very small amount of sample can be obtained from them. BALF EGFR mutation testing does not require paraffin- fixation time of tissue so it can save the time for EGFR mutation testing” was added.

Reviewer 3 Report

In this study by Kim et al, the author present data on the use of extracellular vesicle DNA (EV-DNA) derived from bronchoalveolar lavage fluid (BALF) for EGFR mutation detection and compares its sensitivity, specificity and concordance with EGFR mutation testing from lung tumour tissue and liquid biopsy (cfDNA). 

Due to the greater stability of DNA from EV's offered by the lipid bilayer in comparision to cfDNA which tends to be more fragemented with a shorter half-life, these findings identify the potential use of BALF as a liquid biopsy for EGFR mutations in NSCLC patients. Furthermore, with the shorter turn-around time for EV-DNA processing and subsequent EGFR testing from BALF samples, this may offer a more rapid complementary or alternative approach to both tissue or liquid biopsy (cfDNA).  

Some minor comments for addressing:

[1] Materials and Methods; section 2.2; following isolation and purification of EV-DNA from BALF samples, how was purity and DNA yield quantified? Also, in section 2,4; two kits are alluded to, which were used for the isolation and purification of plasma cfDNA. In reading this part of the text, it is not clear whether one kit (High Pure PCR Template Preparation Kit) was used for the isolation of plasma cfDNA and the other kit (TANBead OptiPure cfDNA Auto Tube) was used for the purification step, specifically? Or was the use of both alternated?

[2] Results; section 3.1; The authors refer to in the incidence of double EGFR mutations in (8) 8.6% of the patient cohort examined, with reference to Table 1. However, if reading the content of this table correctly, and looking at the number of double EGFR mutations, would this not correspond to 6? (ie Exon 19 del+T790M, n=2; G719C+S768I, n=2; L858R+G719C, n=1; L858R+T790M, n=1)

[3] On line 169, give full abbreviation for TBNA (where first mentioned in the text).

[4] Keep abbreviations consistent throughout the manuscript; eg. use BALF liquid biospy or BALiquid.  

[5] Discussion; line 280; replace "tine" with "time"

[6] line 293; insert "exon" into sentence "exon 21 L858R"

Author Response

In this study by Kim et al, the author present data on the use of extracellular vesicle DNA (EV-DNA) derived from bronchoalveolar lavage fluid (BALF) for EGFR mutation detection and compares its sensitivity, specificity and concordance with EGFR mutation testing from lung tumour tissue and liquid biopsy (cfDNA). 

Due to the greater stability of DNA from EV's offered by the lipid bilayer in comparision to cfDNA which tends to be more fragemented with a shorter half-life, these findings identify the potential use of BALF as a liquid biopsy for EGFR mutations in NSCLC patients. Furthermore, with the shorter turn-around time for EV-DNA processing and subsequent EGFR testing from BALF samples, this may offer a more rapid complementary or alternative approach to both tissue or liquid biopsy (cfDNA).  

Some minor comments for addressing:

[1] Materials and Methods; section 2.2; following isolation and purification of EV-DNA from BALF samples, how was purity and DNA yield quantified?

A: We described how purity and DNA yield was quantified in line 121-124. We  added “The quality and length of the purified DNA were analyzed using a 4200 Tapestion and Genomic DNA ScreenTape (Agilent Technologies, Santa Clara, CA, USA). The concentration and purity of DNA samples were measured using the NanoDrop (Thermo Sci-entific, Waltham, MA)”

Also, in section 2,4; two kits are alluded to, which were used for the isolation and purification of plasma cfDNA. In reading this part of the text, it is not clear whether one kit (High Pure PCR Template Preparation Kit ) was used for the isolation of plasma cfDNA and the other kit (TANBead OptiPure cfDNA Auto Tube) was used for the purification step, specifically? Or was the use of both alternated?

A: We used either High Pure PCR Template Preparation Kit or TANBead OptiPure cfDNA Auto Tube. We clarified it in the line 150-151 in the revised manuscript

[2] Results; section 3.1; The authors refer to in the incidence of double EGFR mutations in (8) 8.6% of the patient cohort examined, with reference to Table 1. However, if reading the content of this table correctly, and looking at the number of double EGFR mutations, would this not correspond to 6? (ie Exon 19 del+T790M, n=2; G719C+S768I, n=2; L858R+G719C, n=1; L858R+T790M, n=1)

A: The number 8 in the previous manuscript was incorrect, as the reviewer pointed out. We corrected the mistake, fixing 8 to 6 on line 178. We appreciate the reviewer for careful review.

[3] On line 169, give full abbreviation for TBNA (where first mentioned in the text).

A : We provided the full abbreviation for TBNA on Line 190 where it is first mentioned.

[4] Keep abbreviations consistent throughout the manuscript; eg. use BALF liquid biospy or BALiquid.  

A: We replaced BALiquid to BALF liquid, and BALF liquid is consistently used in the revised manuscript.

[5] Discussion; line 280; replace "tine" with "time"

A: We fixed the typo “tine” with “time” on line 313 in revised manuscript.

[6] line 293; insert "exon" into sentence "exon 21 L858R"

A: We inserted “exon” in line 326 in the revised manuscript.